**Data Availability Statement:** All relevant data are within the manuscript and its Supporting Information files.

**Funding:** This work was supported by MEXT/JSPS KAKENHI Grant, Numbers 17H06380 to KO &

# Sex differences in the development and expression of a preference for familiar vocal signals in songbirds

**Tomoko G. Fujii**[1], **Maki Ikebuchi**[2], **Kazuo Okanoya**[1,2]\*

**1** Department of Life Sciences, Graduate School of Arts and Sciences, The University of Tokyo, Meguro-ku, Tokyo, Japan, **2** Behavior and Cognition Joint Research Laboratory, RIKEN Center for Brain Science, Wako-shi, Saitama, Japan

\* cokanoya@mail.ecc.u-tokyo.ac.jp

## Abstract

Production and perception of birdsong critically depends on early developmental experience. In species where singing is a sexually dimorphic trait, early life song experience may affect later behavior differently between sexes. It is known that both male and female songbirds acquire a life-long memory of early song experience, though its function remains unclear. In this study, we hypothesized that male and female birds express a preference for their fathers' song, but do so differently depending on the developmental stage. We measured preference for their father's song over an unfamiliar one in both male and female Bengalese finches at multiple time points across ontogeny, using phonotaxis and vocal response as indices of preference. We found that in males, selective approach to their father's song decreased as they developed while in females, it remained stable regardless of age. This may correspond to a higher sensitivity to tutor song in young males while they are learning and a retained sensitivity in females because song is a courtship signal that is used throughout life. In addition, throughout development, males vocalized less frequently during presentation of their father's song compared to unfamiliar song, whereas females emitted more calls to their father's song. These findings contribute to a deeper understanding of why songbirds acquire and maintain such a robust song memory.

## Introduction

Early developmental experience can have a large impact on recognition at the species and individual level in various social contexts. Imprinting in precocial birds is a typical example; visual experience soon after hatching enables hatchlings to discriminate parents from other objects during filial imprinting [1,2]. The sight of conspecifics in early life is even critical for sexual preference, which expresses much later in an animal's lifetime [3–5].

Perception and production of more complex courtship signals like birdsong also depends on such developmental experiences. Birdsong is a sequence of species-specific sound elements, acquired through vocal learning [6]. While both sexes sing in many avian species [7–9], singing is a completely sexually dimorphic trait in some species such as zebra finches (*Taeniopygia*

17J07023 to TGF. The funders had no role in study design, data collection and analysis, decision to publish, or preparation of the manuscript. (https://www.jsps.go.jp/index.html)

**Competing interests:** The authors have declared that no competing interests exist.

*guttata*) and Bengalese finches (*Lonchura striata* var. *domestica*), where only males learn to produce songs. Because male juveniles learn song by memorizing and imitating adult song, hearing the songs of conspecifics is necessary for them to develop normal, species-typical song [10].

In these species, both males and non-singing females are able to memorize the song they heard in early life. Previous studies in zebra finches and Bengalese finches have shown that both male and female birds prefer their fathers' song over unfamiliar songs, and that such a preference persists even after they become independent of their parents [11–15]. Early-life experience with their father's song can be an important underpinning of future mate choice in female birds, as some studies have already demonstrated [16–19]. In female zebra finches, for example, developmental song exposure is necessary to evaluate song quality as adults [16–18], and when cross-fostered, birds generalize this preference to songs of the same sub-species as their foster father [19]. Although the function of acquiring and maintaining such a song preference in males is less clear, it may somehow contribute to song learning. Recent studies suggest that tutor-tutee song similarity in male juvenile zebra finches is correlated with the degree of social attachment or attentiveness of a tutee to the song tutor during learning [20,21].

For a deeper understanding of why birds acquire such a long-lasting song memory, further exploration into the ontogeny and sex difference of song preference is necessary. Previous studies that investigated either behavioral or neural response to conspecific songs suggest that researchers need to carefully investigate sex and developmental differences. For example, Bailey et al. [22,23] examined neuronal response to conspecific song in the higher auditory forebrain in zebra finch juveniles by FOS and ZENK expression and found that responses differed between sexes at 30 days post hatch. Multi-unit electrophysiological recordings in the zebra finch auditory forebrain revealed that response selectivity to tutor song is higher in adults than juveniles [24] and that there is a sex difference in response magnitude to tutor song [25]. A study measuring cardiac response to song playbacks in Bengalese finches reported that presentation of novel conspecific song significantly increased the heart rate in females but not in males [26]. In addition, when song discrimination is measured by operant conditioning, the strategy used to perform the task may differ between the sexes. It was suggested that male Bengalese finches rely more on auditory cues when discriminating biological audiovisual stimuli than females do [27]. In another study, the authors conducted a meta-analysis and results suggested that the speed of learning auditory discrimination in zebra finches can be affected by factors including sex, developmental history, and stimulus characteristics [28].

Accordingly, we hypothesized that both male and female birds have a preference for their fathers' song but the way they express this preference can differ depending on both sex and developmental stage. These differences can be elucidated if we carefully choose which behavioral response to measure. In the current study, we aimed to examine this hypothesis by tracking the song preference of male and female Bengalese finches as they developed from juveniles to adults. To this end, the preference assay should be longitudinal and conducted at every important time point of development. Although some previous studies investigated song preference in both sexes and/or repeatedly within an individual [12,13,29], none of these studies were designed to meet this criterion. Among several techniques to quantify song preference, we considered phonotaxis [for example, 12,30] and vocal response [for example, 18,31,32] as these have more ecological validity than an operant assay, which is more suitable for measuring motivation to trigger playback of a particular stimulus [33,34]. Thus, we designed the experiment to test song preference of young birds at multiple time points across development by measuring approach behavior and vocal response to song playbacks.

## Materials and methods

### Animals

The subjects for this study were 10 male and 10 female Bengalese finches from 11 different families in our lab colony. The 11 pairs of adult males and females used for breeding the subjects were either bred in our laboratory or purchased from a pet supplier. All subjects were raised by both parents and housed with their families (parents and other siblings) in a home cage (cage size: 30 × 24 × 33 cm [width × depth × height]) until approximately 120 days post hatch (dph). Each home cage was placed in a colony room but visually separated from one another by opaque partitions. Thus, juveniles could always hear and interact with their fathers, whereas other males in the room could be heard singing but not seen. The number of subjects per brood varied among families; both male and female siblings were recruited from 3 families, and either male(s) or female(s) were recruited from the remaining 8 families (for details see S1 Table). After reaching 120 dph, subjects were introduced to a single-sex group cage (cage size: 37 × 42 × 44 cm) with other birds (Fig 1B). The number of birds kept in a single-sex cage ranged from 8 to 14. To prepare song stimuli, 29 adult males (> 180 dph) were used, which included the fathers of the subjects (11 birds) and 18 other unfamiliar individuals. Prior to the experiments, the subjects had never been exposed to the unfamiliar birds visually or acoustically. These 29 birds were either bred in our lab or purchased from a pet supplier. All birds used in the study were kept under a 14:10 h light:dark cycle with food and water provided ad libitum. In addition, they were fed oyster shell and greens once a week. The temperature and humidity were maintained at around 25 ºC and 60%, respectively. After finishing all the experiments, the subjects, their parents, and the birds used for stimulus recording continued to be housed in our laboratory for additional research. All experimental procedures in this study were approved by the Institutional Animal Care and Use Committee at the University of Tokyo (permission #27–9 and #29–2).

### Song preference test

The preference test was a song playback experiment designed to measure phonotaxis and vocal responses, modelled after previous research [12,14,30] but with some modifications.

**Apparatus.** All preference tests were conducted in a test cage placed in a sound attenuation room (163 × 163 × 215 cm [width × depth × height]). The test cage was a plastic meshed, three-chambered cage (entire size: 105 × 16 × 22 cm; each compartment: 35 × 16 × 22 cm) (Fig 1A). The compartments were divided by a detachable partition, and each one was equipped with two perches. The middle compartment was considered a neutral zone and during the acclimation phase, birds were kept here with the partitions to the other compartments closed. Birds had free access to food and water in this middle compartment during the acclimation and testing phases. During the tests, birds could move freely to any part of the cage. The left and right compartments were considered approach zones, where birds could approach the songs broadcasted from the speakers. Loudspeakers (MM-SPL2N2, SANWA SUPPLY) were located at each end of the test cage and broadcasted song stimuli. Birds' movement and vocal activity during the tests were monitored and recorded via a web camera (BSW200MBK, Buffalo) fixed above the test cage. The frame / sampling rate for video and audio recording were 16 Hz and 44100 Hz, respectively. The light:dark cycle, temperature, and humidity of the testing environment was identical to that of the colony room.

**Stimuli.** Preference for the father's song over an unfamiliar song was tested in both males and females. Within a given testing age, we used songs recorded from a single unfamiliar individual for a given subject but used different singers (unfamiliar individuals) between subjects.

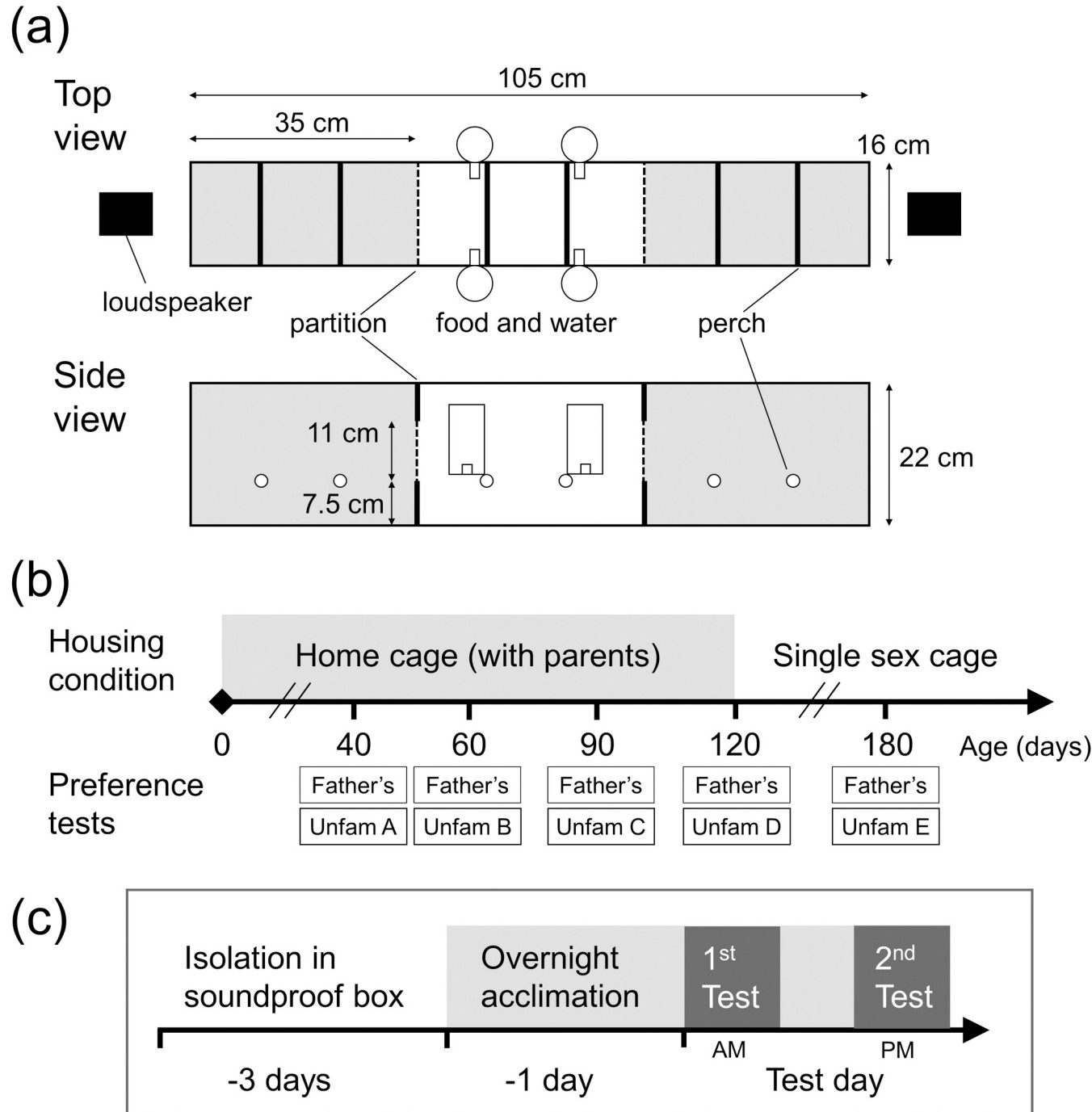

**Fig 1. Overview of experimental methods.** (a) A schematic diagram of the test cage. The upper and lower panels show the top and side view, respectively. Approach zones are indicated with shadings. (b) The overall time course of the experiment. (c) The schedule of a single test at a certain age.

At each age tested, we used a different individual. Thus, a bird was exposed to a new unfamiliar song at each age of testing (Fig 1B). We randomly chose which unfamiliar song to present to each bird at each age to avoid pseudo-replication and minimize any possible bias towards specific songs. In some cases, a father's song for one subject was used as an unfamiliar song for an unrelated subject hatched from another family. This was only done when the breeding period

did not overlap between families and thus the juveniles did not have previous exposure to that song.

Songs were recorded in a soundproof chamber through a microphone (PRO35, Audio-technica) fixed at the top of the cage in the chamber. The microphone output was amplified by a preamplifier (QuadMic, RME) and digitized by an audio interface (Delta66, M-AUDIO) at 16 bits with a 44100 Hz sampling rate. During recording, each bird was isolated in the chamber, which means that their singing was not directed to other individuals. We recorded songs for at least 12 hours for each bird (for sound spectrograms of Bengalese finch songs, see for example [35]). To create song stimuli, we prepared five different renditions of songs for each vocalizer (i.e., the father or an unfamiliar male). Because it was previously reported that Bengalese finches have within-day variations in song features [36], we always chose song renditions from recordings conducted between 9 am to 3pm. We used visual inspection of the sound spectrogram to select a fraction of sound without any background noise caused by birds' movement during recording. The duration of a single song (one song rendition) was 6.94 ± 0.88 ($M \pm s.d.$) seconds. We ensured that a selected song included most of the typical song elements of each singer and this criterion was usually met within a typical 7-second sample. Due to the variation in the number of introductory notes sung (both within and between individuals), we excluded introductory notes from the stimuli. The sound waveform was first band-pass filtered at 500–10000 Hz. We next adjusted the sound amplitude between two songs so that those of matched duration had the same root mean square value. The sound amplitude within an individual (among five renditions) was then normalized by the maximum amplitude value. Finally, the sound volume was determined so that the sound pressure level was 70 dB when measured at a point in the test chamber that was 12.5 cm away from the loudspeaker. We measured the equivalent continuous A-weighted sound pressure level using a sound level meter (NL-27, Rion).

**Testing schedule.** We conducted song preference tests at the following ages, with a range of ± 5 days: 40, 60, 90, 120, and 180 dph (Fig 1B). The variance in testing age came from two sources; 1) there was an error of at most 2 days in identifying the hatching date and/or 2) when we tested more than 1 bird per brood, it was sometimes impossible to test different siblings at the exact same age. Only one male was not tested at 40 dph for technical reasons and his first test was at 60 dph. With the exception of this bird, all other birds were tested at these five time points. The testing ages were determined so that they adequately covered development while also keeping enough time between tests. Bengalese finches are able to feed themselves at around 40 dph and reach sexual maturity at around 120 dph. Birds enter the sensorimotor phase of song learning between 60 and 90 dph, and song structure develops most drastically during this period. Song becomes nearly crystallized at 120 dph but changes are still seen thereafter [35]. We tested whether song preference in juvenile birds is sustained and to what degree after they fully mature at 180dph.

The overall schedule of a single test is as follows (Fig 1C). This procedure was used for all tests. Each bird was isolated in a soundproof chamber three days prior to the test to ensure they were sufficiently motivated to hear songs during testing. In the evening before testing, the bird was introduced to the test chamber for acclimation. The detailed process for acclimation is as follows. First, the subject and his or her mother were put in the middle compartment of the test chamber. Approximately 4 hours later, the mother was taken out from the cage, and the subject stayed overnight in the cage alone. This procedure was based on a finding in our preliminary experiments. We found that introducing a subject to the test chamber on the day of testing without a companion was not sufficient for acclimation. These birds tended to exhibit little locomotion and vocalization, especially during the song playbacks. Mothers were chosen as companions because they do not sing and are highly familiar to the subjects. The

mother stayed in the test cage only during the acclimation phase when songs were never played, and birds were not allowed to enter approach zones. Thus, it is unlikely that her presence biased her offspring's behavior during testing.

A single test was composed of two sessions; one in the morning (at around 9 am) and the other in the afternoon (at around 2 pm). Partitions were removed manually just before the beginning of the session. In the case where a bird moved to either the left or right chamber before the song presentation started, we withheld song presentation and waited until the bird went back to the middle compartment. In each session, the father's song and an unfamiliar song were played back alternately from two speakers. The inter-onset interval was randomly assigned to have mean value of 30 seconds, ranging from 25 to 35 seconds. Each song was played back 20 times in a random order (each of the 5 renditions played back 4 times). Thus, the entire session had 40 song playbacks (20 trials × 2 songs) and took approximately 20 minutes (30 seconds × 40 trials). After the first session, partitions were closed again, and the bird waited in the middle compartment of the cage until the second session began. The presentation order and the speaker position were counterbalanced between the two sessions. For instance, if father's song was played first (odd-numbered trials) from the left speaker in the morning session, it was played second (even-numbered trials) from the right speaker in the afternoon session.

**Behavioral analysis.** We first quantified selective approach to the father's song as in previous studies [12,14,30]. The quantification was performed based on the duration of time that a bird approached the sound source of each song. We measured the duration (in seconds) that a bird stayed in the approach zone of each song. In addition, we measured vocal response (calling in both sexes and singing in males), as some studies have shown that vocal response to playback can be a good indicator of song discrimination or preference [12,29,31,32]. For each stimulus, we counted the number of calls and the number of playbacks (out of 40 playbacks at each age of testing) in which a bird emitted calls or sang specifically during the song presentation. Vocalization during the intervals between song presentations was not measured. Although Bengalese finches have several categories of calls, we included all types of calls emitted during the tests without differentiation in the main analysis. To further examine female vocal response, we analyzed the number of pulse-train like calls (also referred to as distance calls [37]) separately from other types of call, as a previous study reported that the frequency of distance calls emitted during song presentation was positively correlated with the frequency of sexual displays [32]. We did not perform call-type specific analysis on male data, because unlike female calls, it is technically difficult to differentiate male call categories solely based on acoustical features.

For all these types of responses, duration (approach) or frequency (vocalization) of the first (AM) and second (PM) sessions were summed up for each stimulus (father / unfamiliar) at each age of testing and used for later analysis. We examined if there was a significant difference in response patterns between sessions in the morning and afternoon (see Statistical analysis) and confirmed that there was no systematic difference (see S2 Table). Thus, further analysis was conducted with AM and PM data combined. To describe the bias of behavioral responses towards either song at the population level, we calculated the following proportion: the response to father's song divided by the sum of response to both songs. The completely selective response to the father's song or the unfamiliar song results in a value of 1 or 0, respectively. If a bird did not show any response as defined above, it was categorized as 'no response' and excluded from the process of calculating the proportion. The response proportion was only used for descriptive purposes and to examine whether there were response differences between sessions in the morning and afternoon, while actual frequency of behavior (count data) was used for statistical modelling as described below.

## Statistical analysis

We examined whether there are differences in the behavioral response to songs based on sex and developmental stage, using a linear mixed model (LMM) or generalized linear mixed models (GLMMs). In brief, we fitted a model in which Stimulus, Age, and their interaction were specified as explanatory variables, separately to male and female data. Our goal was to compare the results of model estimation between sexes, under the hypothesis that if sex differences contribute to the ontogeny of song preference, then stimulus type, age of testing, and their interaction would predict behavioral response in a different manner depending on sex. We chose this method of analysis rather than directly testing the difference in preference intensity between sexes, because it is possible that sex differences in vocal response may be expressed in a more qualitative manner rather than quantitative. As such, we analyzed the phonotaxis, calling, and singing responses within an identical framework. We additionally analyzed the phonotaxis behavior with a model which includes sex as an explanatory variable. The purpose, statistical procedures and the results are provided in the Supporting Information (S1 Text).

In the analysis of phonotaxis, the response variable was the duration of time each bird spent in the approach zones of two sound sources. We used the logarithmic transformation of the original values so that a linear mixed model could be fitted. Because the data included a duration of zero seconds if a bird showed no entrance into either approach zone, we uniformly added 1 second to all duration data before calculating the logarithmic transformation. With the exception of cases where there was no entrance, the minimum duration of time spent in the approach zone was 5 seconds. Vocal responses were analyzed using GLMMs. In the case of call data, the response variable was the frequency of calls emitted during song presentation. The Poisson distribution was assumed as a probability distribution with a log link function and the number of trials in which each bird vocalized at least one call being set as an offset term. When analyzing male singing behavior, the response variable was the number of trials out of the total 40 trials (playbacks) in which a bird sang. The binomial distribution with a logit link function was specified to represent the probability distribution.

We specified Stimulus, Age, and their interaction as explanatory variables (fixed effects) in all LMM and GLMMs fitted to phonotaxis and vocal response data, respectively. Model fitting was performed for 3 behavioral measures (approach time, call, and song) from males and females separately; thus, there were 5 sets of data (male approach, female approach, male call, female call, and male song). For these 5 datasets, the data acquired at 60 and 120 dph were regarded as behavior representative of pre- and post- sexual maturation, and data acquired at 40, 90, and 180 dph were excluded from this analysis. There were two main reasons for this. First, some birds showed little or no response at 40 dph, which made it difficult to compare this data quantitatively with the data from other ages. Second, statistically comparing all ages of testing with one another increases the number of factor levels and thus the difficulty of model estimation. Instead, we decided to simply focus on the effect of maturation by comparing two ages categorically. The explanatory variable Age was coded as a binary dummy variable, where 0 and 1 were assigned to 60 and 120 dph, respectively. Another explanatory variable, Stimulus, was coded in the same manner; 0 and 1 were assigned to the unfamiliar and father's song, respectively. We assigned bird identity as a random effect, where the slope of regression (Stimulus) was assumed to vary among individuals. LMMs and GLMMs were fitted by restricted maximum likelihood estimation and maximum likelihood estimation (Laplace Approximation), respectively. On estimating the coefficients and their standard errors, $t$-value and $z$-value (Wald statistics) were computed for estimates of LMMs and GLMMs, respectively. Satterthwaite's method was used to estimate denominator degree of freedom to compute $t$-

value. We used these statistics as a reference of whether estimated coefficients were significantly different from zero.

In addition to the LMM/GLMM fitting and description, we conducted two other analyses. First, we tested whether response proportion of each behavioral index significantly differed between sessions in the morning and afternoon (see also Behavioral analysis) in order to verify the procedure of analyzing morning and afternoon sessions combined. For this purpose, we used Mann-Whitney $U$ test to compare the proportion of each behavioral measure between sessions for each sex at each tested age. (S2 Table). Second, to further validate the relevance of call responses, we examined if preference measured by phonotaxis and vocal responses were correlated by calculating Spearman's rank correlation coefficient. Behavioral proportion of one individual at one age represented a single data point. Correlation coefficient was independently calculated for male and female data under the assumption that the behavioral function of calls is potentially different in males and females.

All processes regarding model fitting and statistical testing of estimated coefficients were performed by lme4 package [38] (lmer and glmer functions) and lmerTest package [39] written in R version 4.0.2 [40]. Computation and figure illustration of mean ($M$) and 95% confidence interval ($CI$) of proportion and behavioral frequency data were performed by packages called SciPy (ver. 0.19.0) written in Python. 95% confidence intervals were calculated using the bootstrapping method ($n = 1000$). Mann-Whitney $U$ tests and calculation of Spearman's rank correlation coefficient ($r_s$) were also conducted with SciPy. We set the level of statistical significance at 0.05 for all tests.

## Results

### Song preference measured by phonotaxis

We first measured approach behavior to the sound sources. Birds of both sexes selectively approached their fathers' song over unfamiliar songs (Fig 2A, Table 1), which is consistent

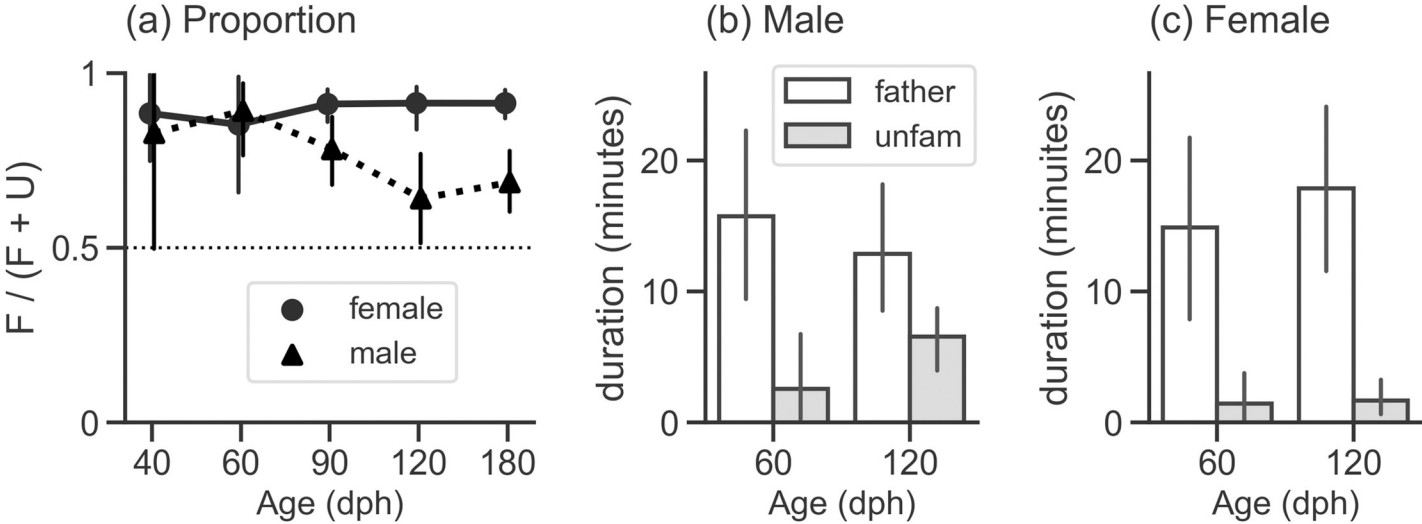

**Fig 2. Results of the preference test (phonotaxis).** (a) Population mean of the response proportion plotted against age of testing (the number of individuals of each sex at each age is specified in Table 1). A solid line with filled circles indicates female data, and a dashed line with filled triangles indicates male data. The proportion was calculated as the duration of time a bird spent in the approach zone of the father's song (F) divided by the total duration of time spent in two approach zones (F + U). (b, c) Population mean of time spent (seconds) in either approach zone in the tests conducted at 60 and 120 dph in males (b) and females (c). We used logarithmic transformation of this time duration data for all individuals (10 males and 10 females) for model fitting. Open bars and grey bars indicate response to the father's song and unfamiliar song, respectively. In all three panels, error bars are 95% confidence intervals. Mean and confidence interval values were summarized in Table 1.

**Table 1. The numerical description of phonotaxis data shown in Fig 2A.**

| Sex | Age (dph) | *mean* | *CI lower* | *CI upper* | *n* |
|---|---|---|---|---|---|
| FEMALE | 40 | 0.885 | 0.746 | 0.995 | 9 |
| | 60 | 0.853 | 0.659 | 0.990 | 9 |
| | 90 | 0.911 | 0.863 | 0.953 | 9 |
| | 120 | 0.914 | 0.842 | 0.961 | 9 |
| | 180 | 0.914 | 0.871 | 0.951 | 10 |
| MALE | 40 | 0.828 | 0.495 | 0.998 | 6 |
| | 60 | 0.892 | 0.769 | 0.973 | 9 |
| | 90 | 0.782 | 0.679 | 0.878 | 10 |
| | 120 | 0.641 | 0.506 | 0.781 | 10 |
| | 180 | 0.687 | 0.601 | 0.772 | 10 |

The columns labelled 'CI lower' and 'CI upper' indicate the lower and upper endpoints of 95% confidence interval, respectively. The number of individuals used to calculate mean proportion and 95% *CI* is shown in the right most column (at each age, birds who did not respond to either song were excluded, see 'Methods–Statistical analysis' for details).

with previous studies reporting preference for the father's song as measured by phonotaxis [11,12,14]. Although the population mean of the behavioral proportion was consistently above 0.5 in both sexes, the value did fluctuate across testing ages. Especially in males, the proportion seemed to decrease after sexual maturation (at 120 and 180 dph).

To examine if the developmental course in song-selective approach differs between sexes, we applied an LMM in which time spent in approach zones was treated as a function of Stimulus (unfamiliar/father's song = 0/1), Age (60/120 dph = 0/1), and their interaction (Fig 2B and 2C). Model fitting was performed on time duration data for males and females independently, with the intent to characterize how these factors predict the behavior of each sex rather than directly comparing the intensity of preference between sexes (for the latter purpose, refer to mean values of proportion with confidence intervals in Fig 2A and Table 1). The result of model estimation is summarized in Table 2. For males, estimated coefficients of Stimulus and Age were significantly above zero, indicating that they stayed longer in proximity of the sound source of father's song and that the time spent in approach zones generally increased with age. Importantly, interaction of Stimulus and Age was also significant, and the coefficient was a negative value, suggesting that phonotaxis specifically to the father's song decreased as males aged. On the other hand, only the

**Table 2. Result of the estimation of a model (LMM) fitted to phonotaxis data.**

| Sex | Variable | Coefficient | Standard Error | df | *t*-value | *p*-value |
|---|---|---|---|---|---|---|
| FEMALE | (Intercept) | 1.096 | 0.299 | 31.950 | 3.659 | < 0.001 |
| | **Stimulus** | 1.342 | 0.424 | 31.950 | 3.168 | 0.003 |
| | Age | 0.549 | 0.340 | 18.000 | 1.616 | 0.123 |
| | Stimulus × Age | -0.2624 | 0.481 | 18.000 | -0.564 | 0.592 |
| MALE | (Intercept) | 1.275 | 0.265 | 28.927 | 4.802 | < 0.001 |
| | **Stimulus** | 1.380 | 0.375 | 28.927 | 3.676 | < 0.001 |
| | **Age** | 1.044 | 0.267 | 18.000 | 3.910 | 0.001 |
| | **Stimulus × Age** | -0.900 | 0.377 | 18.000 | -2.384 | 0.028 |

Estimated coefficients are given with standard error. *t*-values (calculated using Satterthwaite's method) are shown as a reference of whether estimated coefficients are significantly different from zero. Explanatory variables with a *p*-value less than 0.05 are indicated in boldface.

coefficient of Stimulus was significantly above zero for female data. This indicates that unlike males, females preferentially approached their father's song regardless of their age.

## Song preference measured by vocal response

We next analyzed vocal behavior and found that the response was modulated depending on song type but in a different manner between females and males. In females, the proportion of calling frequency was above 0.5 for all ages of testing, meaning that they gave more calls to their fathers' song than to unfamiliar song consistently across development (Fig 3A, Table 3). In contrast, the frequency of call response was not necessarily biased toward either song in males, although the mean proportion was below 0.5 at 90, 120, and 180 dph (Fig 3A, Table 3).

To examine the effect of song type and development on the frequency of call response during stimulus presentation, we analyzed vocal response data using GLMMs (Fig 3B and 3C, Table 4) with a similar rationale and process as for the LMM analysis of phonotaxis. For male data, the only explanatory variable with a coefficient significantly different from zero was Age. This indicates that calling frequency generally increased with age, but that male vocal response was not well predicted by the song type. In contrast, for female data, which included all types of calls, the coefficients of explanatory variables Stimulus and Age were significantly different from zero, although their interaction was not. While the overall amount of calling increased with age, females consistently gave more calls to their father's song over unfamiliar songs at 60 dph and 120 dph. When we applied the same GLMM to female distance call data, the result was similar to the analysis of all types of calls. The coefficient of Stimulus was significantly above zero although the significance for the explanatory variable Age was marginal (Table 4). Interaction of Stimulus and Age did not significantly contribute to predict the response. This result can be regarded as reasonable partly because the major category of calls emitted during song presentation was the distance call; mean ($\pm$ *S.D.*) percentage of distance calls across individuals ($n = 10$) was 74.4 $\pm$ 23.8%. These results are consistent with a previous study [32], which investigated preference among unfamiliar songs and suggests that frequency of distance calls, rather than other categories of calls, can be a good predictor of preference. Given these findings, the distance call can also be an appropriate measure of preference in the current case where preference is tested between the father's song and unfamiliar song in females.

Lastly, we analyzed how song type and age of testing affected male singing behavior during stimulus presentation (Fig 3D and 3E). No birds sang when tested at 40 dph. However, at all other ages, we found that birds sang more frequently when presented unfamiliar songs than when they heard their fathers' song. The proportion of singing frequency was below 0.5 consistently across development (Fig 3D, Table 3). We also fitted a GLMM to the frequency data for male singing response (Fig 3E, Table 4). The estimated coefficients of Stimulus and Age were statistically significant while their interaction was not. This result means that male singing frequency became higher as birds aged, and that the tendency to sing less when presented with their father's song was present before and persisted after sexual maturation.

Although there seemed to be no clear effect of stimulus type on calling behavior in males, singing at all ages was less frequent when they were exposed to their father's song. This stimulus selective vocal response was contrary to that of females, who emitted calls more frequently to their father's song.

## Relationship between phonotaxis and vocal response

We also examined whether song preference measured by phonotaxis and vocal response were correlated. The proportion for phonotaxis and that of vocal response were positively correlated in females ($r_s = 0.638$, $p < 0.001$) but not in males ($r_s = 0.060$, $p = 0.694$). This supports a

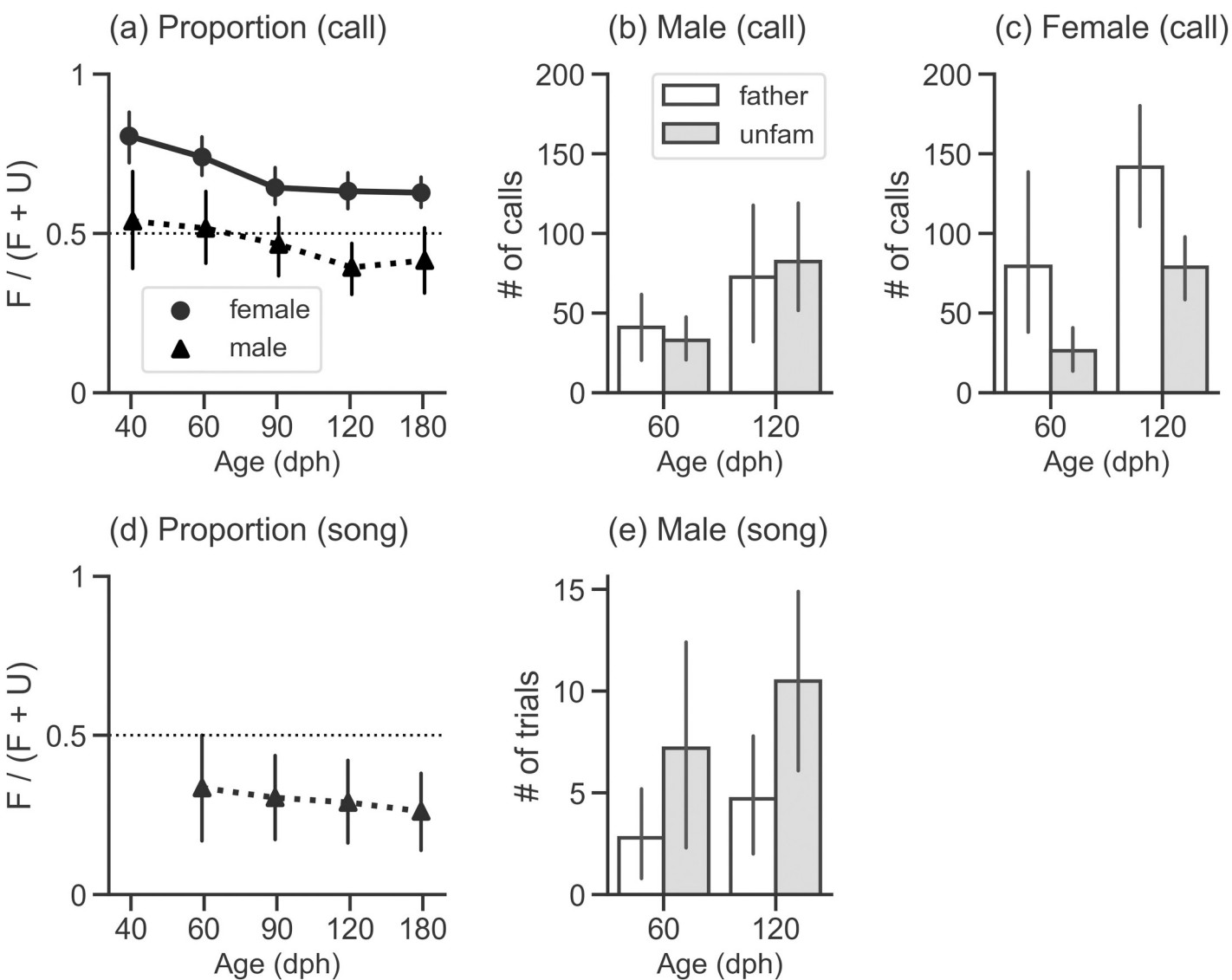

**Fig 3. Results of the preference test (vocal behavior).** (a, d) Population mean of the frequency proportion plotted against age of testing (the number of individuals of each sex at each age is specified in Table 3). (a) shows the results of calling, while (d) shows the results of singing (only males). A solid line with filled circles indicates female data, and a dashed line with filled triangles indicates male data. The proportion was calculated as the response frequency to the father's song (F) divided by the total response frequency (F + U). (b, c, e) Population mean of the number of trials in which birds vocally responded to either song in the tests conducted at 60 and 120 dph. (b) shows the results of male calling, (c) female calling, and (e) male singing. We used this count data for all individuals (10 males and 10 females) for model fitting. Open bars and grey bars indicate response to the father's song and unfamiliar song, respectively. In all three panels, error bars are 95% confidence intervals. Mean and confidence interval values are summarized in Table 3.

possibility that calling response during song presentation may have functional relevance specifically in females but not in males. This result is also consistent with the finding in the GLMM analysis that call frequency was well predicted by stimulus type only in females (Fig 3A, Table 4).

## Discussion

It has been repeatedly shown that songbirds acquire a life-long preference for the song they are exposed to in early life [11–15], though its function remains unclear. In the current study, we

**Table 3. The numerical description vocal response data shown in Fig 3A (call) and 3D (song).**

| Sex/Behavior | Age (dph) | *mean* | *CI lower* | *CI upper* | *n* |
|---|---|---|---|---|---|
| FEMALE Call | 40 | 0.806 | 0.727 | 0.877 | 10 |
| | 60 | 0.740 | 0.683 | 0.801 | 10 |
| | 90 | 0.644 | 0.591 | 0.706 | 10 |
| | 120 | 0.633 | 0.580 | 0.688 | 10 |
| | 180 | 0.628 | 0.582 | 0.676 | 10 |
| MALE Call | 40 | 0.539 | 0.387 | 0.697 | 7 |
| | 60 | 0.516 | 0.391 | 0.621 | 9 |
| | 90 | 0.465 | 0.362 | 0.552 | 10 |
| | 120 | 0.393 | 0.305 | 0.471 | 10 |
| | 180 | 0.415 | 0.316 | 0.517 | 10 |
| MALE Song | 40 | - | - | - | - |
| | 60 | 0.335 | 0.163 | 0.496 | 7 |
| | 90 | 0.304 | 0.178 | 0.444 | 10 |
| | 120 | 0.290 | 0.165 | 0.420 | 10 |
| | 180 | 0.262 | 0.140 | 0.379 | 10 |

The columns labelled 'CI lower' and 'CI upper' indicate the lower and upper endpoints of 95% confidence interval, respectively. The number of individuals used to calculate mean proportion and 95% CI is shown in the right most column (at each age, birds who did not respond to either song were excluded, see 'Methods–Statistical analysis' for details).

hypothesized that sex and developmental stage affect how song preference is expressed. This hypothesis is based on the sexual dimorphism of singing behavior, as well as previous research on the effects of sex and developmental stage on neural activity and behavioral discrimination of song stimuli [22–28]. We tested this hypothesis by a series of song playback experiments

**Table 4. Results of the estimation of models (GLMMs) fitted to vocal response data.**

| Sex/Behavior | Variable | Coefficient | Standard Error | *z*-value | Wald *p*-value |
|---|---|---|---|---|---|
| FEMALE all call types | (Intercept) | 0.758 | 0.113 | 6.734 | < 0.001 |
| | **Stimulus** | 0.483 | 0.150 | 3.214 | 0.001 |
| | **Age** | 0.147 | 0.073 | 2.011 | 0.044 |
| | Stimulus × Age | -0.105 | 0.086 | -1.223 | 0.221 |
| FEMALE distance call | (Intercept) | 0.550 | 0.156 | 3.526 | < 0.001 |
| | **Stimulus** | 0.545 | 0.203 | 2.688 | 0.007 |
| | Age | 0.198 | 0.105 | 1.890 | 0.059 |
| | Stimulus × Age | -0.127 | 0.117 | -1.090 | 0.276 |
| MALE all call types | (Intercept) | 0.757 | 0.119 | 6.339 | < 0.001 |
| | Stimulus | 0.177 | 0.168 | 1.050 | 0.294 |
| | **Age** | 0.246 | 0.068 | 3.606 | < 0.001 |
| | Stimulus × Age | -0.130 | 0.095 | -1.370 | 0.171 |
| MALE Song | (Intercept) | 1.589 | 0.325 | 4.888 | < 0.001 |
| | **Stimulus** | -0.955 | 0.486 | -1.965 | 0.049 |
| | **Age** | 0.377 | 0.151 | 2.494 | 0.013 |
| | Stimulus × Age | 0.141 | 0.278 | 0.505 | 0.614 |

Estimated coefficients are given with standard error. Wald statistics (*z*-value) are shown as a reference of whether estimated coefficients are significantly different from zero. Explanatory variables with a p-value less than 0.05 are indicated in boldface.

where a bird was exposed to its' father's song and an unfamiliar song, alternately. Phonotaxis as well as vocalizations were recorded and analyzed as behavioral indices of song preference.

## Phonotaxis behavior and sensory learning in males

Firstly, we found that preference for father's song measured by stimulus selective approach decreased with age in males (Fig 2), which suggests that males were more attentive or attracted to their father's song when young but less so as they matured. Because juvenile males need to hear and memorize tutor song in the process of vocal learning, it is possible that the degree to which juveniles are inclined to listen to tutor song is related to the sensory learning process. The results of recent studies in zebra finches focusing on the social interaction between adult song tutors and juveniles are concordant with this idea. For instance, juveniles who paid more attention to a singing tutor imitated the song more accurately [20]. In another study, authors measured the proportion of time spent in proximity to parents and other conspecifics in a behavioral test as an index of social motivation in juveniles. They reported a positive correlation between this social motivation index and tutor-tutee song similarity [21]. Moreover, when multiple tutors were available, male juvenile zebra finches were more likely to incorporate song elements from their fathers or adults with whom they had stronger social bonds than other conspecific males [41,42].

To determine the relationship between the sensory experience in tutor-tutee interaction, preference, and song template formation, it needs to be directly tested whether higher preference for tutor song results in more successful acquisition of the sensory template. So far, a few studies have examined the correlation between song preference and song learning performance in male zebra finches, but one study found no significant correlation [43] and the other study had mixed results [44]. Because both studies tested song preference in adult birds, future work needs to measure preference in young birds and after manipulating developmental experience.

## Interpretation of stimulus-specific singing response in males

The decrease in selective approach to their father's song does not mean that males became insensitive to the stimulus difference, since frequency of singing was lower during the presentation of their father's song consistently across development (Fig 3D and 3E). There are two possible interpretations for why birds may sing during song presentation in this experiment. Singing behavior could be due to habituation or lack of attention, or it could be due to recognition of the song as a potential competitor. When young, males may suppress singing during presentation of their father's song because they are listening to the song more carefully [20]. As they become sexually mature, they might sing more actively to an unfamiliar song in the context of male-male competition. Although our results cannot provide evidence for the proximate or ultimate reason that male Bengalese finches sing in this experimental context, we believe that the relative rate of singing in response to song playback can be used as a measure of song memory in male songbirds.

## Why females prefer their father's song

In contrast to males, both phonotaxis and call responses were better predicted by song type rather than age in females (Figs 2A, 2A, 3A and 3C). This result suggests that the father's song remained relatively attractive even after females matured. It is not clear, however, whether females were sexually attracted to their father's song, or simply attracted due to stimulus familiarity. Preference as measured by call response might partially answer this question. We found that females emitted more calls to their father's song than to unfamiliar song (Fig 3A and 3C),

and previous studies on Bengalese finches [32] and other species [18,31] revealed that frequency of calling can be a reliable indicator of song preference. On the other hand, our experiments in addition to other studies [12,29,45] showed that birds that have not yet reached sexual maturity also responded with more calls to songs they heard in early life. Additionally, assortative mating with a male who sings the same or a similar song to their fathers' can be a successful strategy in some species, while choosing a different song is a better way to avoid inbreeding in other species, depending on the ecological conditions [46]. Thus, future work to learn how the father's song is actually recognized in adult female Bengalese finches should include teasing apart the familiarity and sexual attractiveness of the songs as well as measuring behaviors that are more functionally relevant to reproduction, such as copulation solicitation display [47,48]. Careful investigation into the relationship between other types of behavioral responses [for example, see 49] might also be an effective approach, similar to our current attempt to examine the correlation between phonotaxis and call response measures.

## Methodological issues

In the current study, we chose approach and vocal response as behavioral indices, aiming to capture how birds react to their father's song and unfamiliar songs. We demonstrated that even though both male and female birds retained the memory of their fathers' song, they expressed that memory with different behaviors. Although we have discussed the functional interpretation of this sex difference, it should also be considered from a more methodological perspective. For example, the decrease in song selective approach only in males could have resulted from a change in the ecological validity of the assay, rather than a difference in the functional meaning of song. In other words, testing song preference by phonotaxis may have less ecological validity in males compared to adult females [47] or juveniles. This is because songs are presumably attractive to females or juveniles [reviewed in 50], but more relevant to competition in adult males [6].

Also, we need to use caution when comparing results of experiments that use different types of testing and methodologies. Although some studies demonstrated consistent results using multiple preference indices at the group mean level [48,49], few studies showed consistency between results of different tests at the individual level [32]. A previous study used an operant task to investigate preference for the father's song in adult male and female zebra finches, and reported no significant sex differences [13]. This is inconsistent with our finding that a decrease in stimulus selective phonotaxis was only the case for males. Of course, this difference could be due to a difference in the species tested. Regardless, birds may not express their preference in the same manner in different situations; active control over song playbacks and passive response to playbacks do not necessarily depend on the same neural mechanisms. Finally, while we tested our subjects at multiple time points along development and examined whether the age of testing predicts birds' behavior to songs separately for males and females, Riebel et al. [13] carried out the preference test only once in adults and directly compared males and females. A longitudinal experimental design like ours inevitably introduces the confounding factor of repeated experience with the testing situation, which could be confused with developmental change [51]. Repeated testing can result in the development of responsiveness to songs which differs from that of birds that do not undergo repeated testing. Likewise, experience with a test can affect the response of the animal in the next test. In our study, the former case would be less likely because the relative amount of song exposure during tests is relatively small compared to song exposure in the aviary when birds are not being tested. Theoretically and ideally, the latter case could be avoided by testing birds at only one age (either at 40, 60, 90, 120, or 180 dph) and comparing these results to the data obtained with longitudinal

testing. However, this strategy was practically difficult due to time and space restrictions. Thus, it is especially important when interpreting ontogeny of responsiveness to keep in mind that the results might also depend on the accumulated experience of testing.

## Neural mechanisms underlying song memory

Finally, integrating the results of behavioral and neurobiological studies would be helpful for a deeper understanding of song preference. Accumulating neurobiological evidence suggests that the caudomedial nidopallium (NCM) and caudal mesopallium (CM), the avian higher auditory forebrain, are important for memorization and storage of early song experience [52–54]. A line of studies investigated neuronal responses of these higher auditory areas to tutor (father's) song in adult zebra finches by measuring protein products of immediate early genes. They found that exposure to the father's song compared to unfamiliar song led to a significantly larger number of Zenk-immunopositive cells in the female caudomedial mesopallium (CMM), whereas there was no such stimulus-specific expression in the male CMM [43,52,55]. Other electrophysiological studies demonstrated that there is a population of NCM neurons that selectively respond to tutor song and that the emergence of such a response property actually depends on auditory experience [56,57]. Also, neurotoxic lesions of the NCM resulted in a decrease in tutor song preference (measured by selective approach behavior) in adult male zebra finches [14]. Similarly, it was found that CM lesions in adult female zebra finches and Bengalese finches altered the expression of song preference [58,59]. In the future, studying the correlation between behavior and neural activity as well as how behavior changes as these brain areas are manipulated will serve to more accurately interpret previously reported song-related preference behaviors.

## Supporting information

**S1 Table. Additional information about the subjects.** The number of subjects of each sex per brood is summarized.
(XLSX)

**S2 Table. Results of statistical tests on difference between sessions in the morning and afternoon.** The mean proportion of each behavioral index and standard deviation were calculated for sessions in the morning (am) and afternoon (pm). The number of individuals used to calculate mean proportion is shown in the columns '$n$ (am/pm)' (at each age, birds who did not respond to either song were excluded). $U$ and $p$ indicate the statistical value and $p$-value of Mann-Whitney $U$ tests.
(XLSX)

**S3 Table. Raw behavioral measures of preference tests.** Duration and/or frequency of all behaviors (approach, call, and song) measured in the preference tests of both sexes at all ages are summarized.
(XLSX)

**S1 Text. Additional analysis of song-selective approach responses.**
(DOCX)

## Acknowledgments

We are grateful to Dr. Beth A. Vernaleo for her careful proof reading of this paper.

## Author Contributions

**Conceptualization:** Tomoko G. Fujii, Kazuo Okanoya.

**Data curation:** Tomoko G. Fujii.

**Formal analysis:** Tomoko G. Fujii, Maki Ikebuchi.

**Funding acquisition:** Tomoko G. Fujii, Kazuo Okanoya.

**Methodology:** Tomoko G. Fujii, Maki Ikebuchi.

**Resources:** Kazuo Okanoya.

**Supervision:** Kazuo Okanoya.

**Visualization:** Tomoko G. Fujii.

**Writing – original draft:** Tomoko G. Fujii.

**Writing – review & editing:** Maki Ikebuchi, Kazuo Okanoya.

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
