## [Decision Letter · Decision Letter 0]

5 Aug 2020

PONE-D-20-07853

Sex differences in the development and expression of a preference for familiar vocal signals in songbirds

PLOS ONE

Dear Dr. Fujii,

Thank you for submitting your manuscript to PLOS ONE. After careful consideration, we feel that it has merit but does not fully meet PLOS ONE’s publication criteria as it currently stands. Therefore, we invite you to submit a revised version of the manuscript that addresses the points raised during the review process.

Your study is interesting and tackles an important question in song ontogeny. However, the rationale behind the choice of this specific experimental design and the associated statistical analysis (for example to have separate analysis for male and female) needs better justifying and details. You need to discuss tha potential limitations of your study because of this or think of ways to reach your initial objective. Accordingly, a thorough revision of the methods and results section is expected, and if not possible to detail enough tor each the initial purpose, to discuss the results with aditional care and consideration for the potential confounding effects

We look forward to receiving your revised manuscript.

Kind regards,

Nicolas Chaline

Academic Editor

PLOS ONE

Journal Requirements:

2. In your Methods section, please provide additional details regarding all birds used in your study and ensure you have described the source. For more information regarding PLOS' policy on materials sharing and reporting, see https://journals.plos.org/plosone/s/materials-and-software-sharing#loc-sharing-materials.

3. In your Methods section, please include a comment about the state of the animals following this research. Were they housed for use in further research?

4. Please upload a copy of Supporting Information Table S1 which you refer to in your text on page 31.

Reviewers' comments:

Reviewer's Responses to Questions

**Comments to the Author**

1. Is the manuscript technically sound, and do the data support the conclusions?

Reviewer #1: Yes

Reviewer #2: Partly

2. Has the statistical analysis been performed appropriately and rigorously? 

Reviewer #1: Yes

Reviewer #2: No

3. Have the authors made all data underlying the findings in their manuscript fully available?

Reviewer #1: Yes

Reviewer #2: No

4. Is the manuscript presented in an intelligible fashion and written in standard English?

Reviewer #1: Yes

Reviewer #2: Yes

5. Review Comments to the Author

Reviewer #1: This was an interesting and well done developmental paper that looked at behavioural responses to familiar and unfamiliar songs across significant developmental time points. I do have some comments, questions, and suggestions for clarification.

Line 121: How did the authors select the different song renditions? At random? Or was there some criteria?

Line 123-124: How was a typical song determined? More details are needed.

Line 124: Why were introductory notes excluded. Would this not render the songs unnatural sounding?

Line 174: Did the authors explore the possibility that there were systematic differences with what call types were emitted based on the caller and the stimuli presented? This might make sense to explore this question as it might yield interesting results.

Line 175: Similar to the previous point, did the authors examine am and pm responses separately? There may be interesting patterns here.

Lines 225-227: I do not think the authors wanted this first sentence bolded.

Reviewer #2: Vocal learning in songbirds is well documented, but research has focussed mostly on song production learning. In this study the authors present an interesting study following the development of preference for early tutor song in male and female Bengalese finches. Bengalese finches are an important model for song development studies and a closer look at the trajectory for song perception learning has not been conducted previously. A strong point of this study is that males and females are tested the same way and at the same dates during development.

There is substantial theoretical interest in better documentation of individual and sex differences in song learning strategies (Beecher & Brenowitz 2005, Riebel et al. 2019) so this is a welcome contribution.

I have a number of questions/suggestions regarding the analyses, but these should pose hopefully no problems to address:

1) The authors use a standard phonotaxis approach that has been used in this and other species before to measure song preferences, however, other than most studies they chose as main response variable is ‘entrances’ to the approach zones rather than time spent, see e.g. in the zebra finch (e.g.Ten Cate et al. 1984, Clayton 1988, Witte 2006). Both are informative of the interest the bird has in a particular stimulus but in many species time spent, rather than the number of approaches has proven to be the better predictor of preference (or a combined measure – namely the mean duration per visit). This has two reasons: a very strong preference could mean approach and staying in the preferred approach zone – this behaviour would yield a very low visit score. Second, the entrance/visit variable will become terribly inflated if birds become agitated/aroused by songs of great salience – in a cage as small as yours, a bird reacting with increased motor activity is bound to enter both zones in quick succession when starting to fly back and forth between the main perches. Could you report how the duration of the visits compares to the number of visits? If they are highly correlated using just the one parameter is fine, but if the duration of the visits (i.e. time spent) is not correlated with the number of visits than this parameter is likely to show you a different dimension of the response. Likewise, it would be interesting to know whether the number of calls also covary with one or both approach responses? This would also give you some validation of the call measure with respect (to the already established) approach measure?

2) The aim of the study is to compare male and female behaviour and the authors have provided a good example of how to test males and females in the same way, thereby avoiding that testing differences could consist a confound that could be mistaken as a sex difference. However, they chose to analyses the male and female data in separate analyses – this is surprising and has not been explained. Male and female data that are currently analysed in separate models should be analysed in the same models with sex as a main factor, these analyses should be reported too.

Another important point for the analyses I noted: The design consists of the same tests for 10 males and 10 females from 11 different broods. This suggests that there is a male and a female per tutor? If so the design would be much stronger (consisting of paired m/f data belonging to the same father?) In this case tutor ID should be part of the analyses.

3) A problem with experimental studies of behavioral development involving repeated testing is that age effects can be difficult to disentangle from learning within the experiment. This is a dilemma for studies of behavioral development. It can only be overcome by having many experimental groups each of which gets only tested at a single age and to compare these data to data from groups that get tested at all ages. I realise that for a species with long development and separate housing required in song learning studies this is not always logistically possible but it is important to acknowledge and discuss that some of the changes could potentially have arisen because the birds have experienced the testing setup up repeatedly.

Specific remarks per line number

34-38 at this stage in the intro no taxonomic scope has been given – clearly not all species show learned recognition and not all species have hatchlings –

49 these refs are all zf and bf – perhaps to indicate that female preference learning is a more general phenomenon beyond the estrilid finches, perhaps quote a review here to stress this as a general phenomenon in females? And make readers more aware that looking at this phenomenon in males is much rarer and unresolved?

59ff The referencing in this section and argument is too sweeping and confusing. Your sentence suggests that all these studies have found age and sex differences whereas quite a few did not – especially in zf behavioural response to tutor song is not necessarily different in males and females e.g. (Clayton 1988, Riebel et al. 2002) whereas Kriengwatana et al (2016)presents a metaanalyses of studies on discrimination learning involving artificial stimuli and artificial grammar – in their study females performed slightly better but confer this to another type of discrimination tasks where males were reported to perform better - (Cynx & Nottebohm 1992) which suggest that outcomes of such study might not have a general m/f pattern but much might depend on the type of test? For this paragraph it would thus help readers if you were more specific about what the studies you quote show (see also cmts 49) line 59-60 should also state what the response will be about – the responses to tutor song are not necessarily predictive of other stimuli and vice versa - so please be specific about what questions these studies investigated.

61-63 this is interesting as the only species I am aware of where this has been tested is the zebra finch and here males and females prefer the tutor song (Clayton 1988, Riebel et al 2002)

81, 83 give cage sizes, day length food etc. – see ARRIVE guidelines (seeKilkenny et al. 2010)

83 with how many other birds?

86 “ another 29 adult males” the ‘another’ is confusing as so far there has been no mentioning of adult subjects (only 10m/f offspring) and if the fathers are in the sample it is also confusing – better specify how many of these males were the fathers (I suppose 11?) It is also unclear why you use the many extra males? You would need one unfamiliar song per tutor = 11 unfamiliar (I am assuming males/females from the same tutor also get the same unfamiliar song, please specify this). Ideally you would use the other tutors – as this allows checking whether any songs are very attractive in themselves – and there would be no need to have extra males? Please give more detail here how the stimulus pairs were made and whether a brother/sister pair got the same stimuli.

97 phonotaxis test?

129 give the settings of the sound meter

150 putting the mother in as company seems wise from a welfare point – but can you be sure that she isn’t also reacting to the songs and thus guiding their offspring’s responses?

155 was this always the same unfamiliar song or different ones?

122 Please clarify if this duration for one song or for the song stimulus (consisting of 5 renditions of the song?)

130 would that be the same amplitude a bird sitting at 12,5 cm away from another bird would experience?

144 at 40 dph if you put one subject in, what happened to other siblings – did they stay with the father? If the birds were acclimated for 3 days the rest of the brood would get older (or were they not tested? sample size suggests not but please state specifically if that only one male/female were tested per brood?)

- Can you also specifically mention whether this procedure was the same for the later trials (3 days acclim + mother before testing)?

158 just say 40 playbacks (as being moved in the test chamber once might mean ‘trial’ for most readers, the playbacks are stimulus exposures during a trial)

168 most studies measure the time spent rather than the entries (a bird could only enter once and stay there which would be a very strong preference, whereas a very agitated bird could fly back and forth a lot and would have many entrances?) – please comment on why you chose entrances rather than time spent and whether the two measures were correlated or not correlated (in which case they would measure different dimensions of the response?)

178-180 this is a proportion not a ratio

193 and 252 what was the rationale to test males and females separately – your study explicitly sets out to test for sex differences – so the data of males and females have to be in the same model – as is a differences in df and also different models parameters (you are not suing the same final models for males and females as table 1 suggests where bold face AIC is not always for the same model?)

228-234 present these data in a table to improve readability of the text and to aid better visual comparison of the many values?

References

- Please be consistent in how you use caps in titles

- species names should be in italics

Beecher MD and Brenowitz EA 2005: Functional aspects of song learning in songbirds. Trends Ecol Evol 20: 143-149. 10.1016/j.tree.2005.01.004

Clayton NS 1988: Song discrimination learning in zebra finches. Anim Behav 36: 1016-1024.

Cynx J and Nottebohm F 1992: Role of gender, season, and familiarity in discrimination of conspecific song by zebra finches (taeniopygia guttata). Proc Natl Acad Sci USA 89: 1368-1371.

Kilkenny C et al. 2010: Improving bioscience research reporting: The arrive guidelines for reporting animal research. PLoS Biol 8: e1000412.

Kriengwatana B et al. 2016: Auditory discrimination learning in zebra finches: Effects of sex, early life conditions and stimulus characteristics. Anim Behav 116: 99-112. 10.1016/j.anbehav.2016.03.028

Riebel K et al. 2019: New insights from female bird song: Towards an integrated approach to studying male and female communication roles. Biol Lett 15: 20190059. 10.1098/rsbl.2019.0059

Riebel K et al. 2002: Sexual equality in zebra finch song preference: Evidence for a dissociation between song recognition and production learning. Proceedings of the Royal Society of London Series B - Biological Sciences 269: 729-733.

Ten Cate C et al. 1984: The influence of differences in social experience on the development of species recognition in zebra finch males. Anim Behav 32: 852-860.

Witte K 2006: Time spent with a male is a good indicator of mate preference in female zebra finches. Ethology Ecology & Evolution 18: 195-204.

6. PLOS authors have the option to publish the peer review history of their article (what does this mean?). If published, this will include your full peer review and any attached files.

Reviewer #1: No

Reviewer #2: No

---

## [Author Response · Author response to Decision Letter 0]

7 Sep 2020

We appreciate the reviewers’ and editor’s thoughtful and helpful comments on our manuscript. We realized the problems raised by the reviewers especially regarding the logic behind the method of analysis. Thus, we modified our choice of behavioral index and statistical tests and provided a clearer explanation of why we adopted this method. We also revised the discussion section to consider potential limitations. Details about modification of the analysis as well as specific responses to each comment from the editor/reviewers are described below. Original comments and questions from the editor/reviewers are italicized. All page and line numbers we used to specify the location of changes refer to the unmarked version (clean copy) of our revised manuscript unless otherwise mentioned.

Editor:

> The rationale behind the choice of this specific experimental design and the associated statistical analysis (for example to have separate analysis for male and female) needs better justifying and details. You need to discuss the potential limitations of your study because of this or think of ways to reach your initial objective. Accordingly, a thorough revision of the methods and results section is expected, and if not possible to detail enough to reach the initial purpose, to discuss the results with additional care and consideration for the potential confounding effects.

 Thank you for your comment. As you pointed out, the original version of the manuscript lacked a convincing explanation for the choice of experimental design and the associated analysis. Our aim was to characterize how song type (father’s and unfamiliar), age, and their interaction predict the behavior of each sex, under the hypothesis that if sex differences contribute to the ontogeny of song preference, then stimulus, age of testing, and their interaction would predict behavioral response in a different manner depending on sex (p. 14, lines 236-242). We modified the choice of behavioral index and statistical tests and explained these points in the Methods (p. 12, lines 204-206; pp. 14-16, lines 243-284) and Results (p. 19, lines 321-325) sections. Please also see our response to the similar comment by reviewer 2 (on page 6 of this letter). In addition, we added a discussion on the limitation of the current experimental design a note of caution when interpreting the results (p. 30, lines 495-506).

> 1. Please ensure that your manuscript meets PLOS ONE's style requirements, including those for file naming.

> 2. In your Methods section, please provide additional details regarding all birds used in your study and ensure you have described the source.

> 3. In your Methods section, please include a comment about the state of the animals following this research. Were they housed for use in further research?

>4. Please upload a copy of Supporting Information Table S1 which you refer to in your text on page 31.

We modified the format and style to meet PLOS ONE’s journal requirements (1). We also provided the state of the animals following this research and additional information about all birds we used in the study (p. 7, lines 106-108), which was partly lacking in the original manuscript (2, 3). We uploaded Supporting Information S1-S3 Tables (4).

Reviewer 1:

> Line 121: How did the authors select the different song renditions? At random? Or was there some criteria?

Because it was previously reported that there are within-day variations in acoustic and temporal features of songs in Bengalese finches (Tachibana et al., 2015). we always chose song renditions from recordings conducted between 9 am to 3pm. We chose the renditions evenly within this time period so that there was a comparable number from the morning and the afternoon, and samples were taken throughout the morning and afternoon periods. We described this point on p. 9 in lines 148-150.

> Line 123-124: How was a typical song determined? More details are needed.

We selected fractions of recorded songs so that each fraction included most of the song elements in that singer’s repertoire. Using visual inspection of the sound spectrogram, we found that choosing a fraction of approximately 7 seconds (on average) was adequate to meet this criterion. As the description in the original manuscript may have not sufficiently conveyed this, we tried to elaborate the text to clearly show how we created song stimuli from the original recording data (p. 9, lines 150-154).

> Line 124: Why were introductory notes excluded. Would this not render the songs unnatural sounding?

We followed the procedure of excluding introductory notes, as was done in a previous study with the same species (Dunning et al., 2014). Because there is within- and between-individual variation in the number of introductory notes sung, we decided to exclude them rather than choosing a song rendition with a specific number of notes or editing the number of introductory notes. We explained this point in the manuscript (p. 9, lines 154-155). It unlikely that songs without introductory notes sounded unnatural to birds, as subjects responded appropriately to stimuli by approaching and vocalizing during testing.

> Line 174: Did the authors explore the possibility that there were systematic differences with what call types were emitted based on the caller and the stimuli presented? This might make sense to explore this question as it might yield interesting results.

Thank you for your suggestion. We additionally analyzed distance calls (characterized by a pulse-train like sound) in females separately from other call types, as this category of calls was previously used as an index of song preference in the Bengalese finch (Dunning et al., 2014). We described why we chose this call type and how we categorized the calls in the Methods section (pp. 12-13, lines 211-218) and reported the results (p. 23, lines 373-383, Table 4). To our knowledge, there has been no report that a specific call type is relevant when male birds respond to songs. Also, with the quality of current recording data, it is technically difficult to reliably differentiate call types in males based on acoustic features. Thus, we conducted this additional analysis for females but not for males.

> Line 175: Similar to the previous point, did the authors examine am and pm responses separately? There may be interesting patterns here.

We had not analyzed am and pm responses separately, and thus performed an additional analysis to see whether birds showed different patterns of behavior between the morning and afternoon. Briefly, we calculated the proportion of behavior (response to father’s song divided by the total response to both songs) for am and pm tests separately and compared them for each sex at each age. As we did not find systematic differences between am and pm, we kept our main analysis with am and pm data combined. However, we described this statistical procedure in the Methods section (p. 16, lines 275-280) and reported the results as a part of the Supporting Information (S2 Table) because the time of day can be an important factor, as you pointed out, and readers may be interested in the analysis.

> Lines 225-227: I do not think the authors wanted this first sentence bolded.

We bolded the sentence as we initially intended for it to be a heading. However, after some thought we decided this was too lengthy as a heading and modified it. (p. 17, line 294).

Reviewer 2:

1) The authors use a standard phonotaxis approach that has been used in this and other species before to measure song preferences, however, other than most studies they chose as main response variable is ‘entrances’ to the approach zones rather than time spent, see e.g. in the zebra finch (e.g.Ten Cate et al. 1984, Clayton 1988, Witte 2006). Both are informative of the interest the bird has in a particular stimulus but in many species time spent, rather than the number of approaches has proven to be the better predictor of preference (or a combined measure – namely the mean duration per visit). This has two reasons: a very strong preference could mean approach and staying in the preferred approach zone – this behaviour would yield a very low visit score. Second, the entrance/visit variable will become terribly inflated if birds become agitated/aroused by songs of great salience – in a cage as small as yours, a bird reacting with increased motor activity is bound to enter both zones in quick succession when starting to fly back and forth between the main perches. Could you report how the duration of the visits compares to the number of visits? If they are highly correlated using just the one parameter is fine, but if the duration of the visits (i.e. time spent) is not correlated with the number of visits than this parameter is likely to show you a different dimension of the response.

Thank you very much for bringing this important point to our attention. We originally measured both duration and frequency of phonotaxis but chose visiting frequency as an index for technical reasons, particularly regarding the statistical analysis: 1) in the design of the current experiment, time spent in approach zones of father’s song and unfamiliar song are not independent from one other, 2) partly due to the first point, it is difficult to assume what kind of probability distribution underlies the time duration and thus we could not suppose an appropriate method to statistically model the time duration data, and 3) we have confirmed that preference (behavioral proportion) measured by time spent and visiting frequency were positively correlated (male: rs = 0.849, p < 0.001; female: rs = 0.881; p < 0.001; please also see the scatter plot Fig R1 below) and thought that visiting frequency (count data) could be used instead of time spent.

Fig R1. Plotted behavioral proportion (response to father’s song divided by total response) of time spent on x axis and entrance on y axis. One data point indicates a test at certain age of an individual (thus the number of data points equals to #individuals * 5 (tests at 40, 60, 90, 120, & 180 days old)) Fig R2. We calculated mean time duration a bird spent in each approach zone per one visit then visualized the result at each age of testing as boxplots with female and male data pooled (one data point indicates one individual at each age). Light and dark grey boxes indicate responses to father’s and unfamiliar song, respectively. Y-axis was log-scaled to increase visibility. 

Aside from those technical/practical issues, however, the time duration that a bird spent in proximity to the stimulus would be the better predictor of preference than the number of approaches for the reasons you raised. While revising the manuscript we also calculated the mean time duration per visit and found that birds stayed longer in the zone of father’s song than unfamiliar song on average (Fig R2). This result is predictable and consistent with the fact that the behavioral proportion calculated for time spent is usually slightly higher than the proportion calculated for visiting frequency as seen in Fig R1. Thus, we thought that analyzing phonotaxis with visiting frequency could underestimate the intensity of preference, which may be misleading and make it difficult for readers to compare the current results to previous findings. We therefore changed the behavioral index to time spent and used the logarithmic transformed value to fit linear mixed models to the data (pp. 12, lines 204-206; p. 14, lines 243-248). Accordingly, we revised Fig 2, which originally described visiting frequency data. However, this change of behavioral index did not alter the results we reported; while males decreased their response selectiveness to father’s song as they aged, the preference was stable across age in females. You will find this point if you compare the previous and new versions of Fig 2a. We included Figs R1 and R2 in this letter but not in the revised manuscript, as we felt this information was not necessary for most readers.

On the issue of choosing an appropriate behavioral measure, we also changed the analysis of call responses (please see p. 12, lines 208-210; p. 15, lines 249-252, and Fig 3a-c). Although the original analysis was based on the number of trials in which a bird emitted calls (any number), we now use the actual frequency of calls emitted. Similar to the analysis of phonotaxis, our main finding remained the same, even with this modification.

(reviewer’s comment continued) Likewise, it would be interesting to know whether the number of calls also covary with one or both approach responses? This would also give you some validation of the call measure with respect (to the already established) approach measure?

Thank you for your helpful comment. We checked if the proportion (intensity of preference to father’s song) of time spent and number of calls were correlated. We found that these were positively correlated in females (rs = 0.638, p < 0.001) but not in males (rs = 0.060, p = 0.694), which suggests that call number may be a valid measure for females but not for males. We described how we analyzed the correlation on p. 16, lines 280-284 and reported the results on p. 25, lines 403-409, and added related discussion on pp. 28-29, lines 468-471.

2) The aim of the study is to compare male and female behaviour and the authors have provided a good example of how to test males and females in the same way, thereby avoiding that testing differences could consist a confound that could be mistaken as a sex difference. However, they chose to analyses the male and female data in separate analyses – this is surprising and has not been explained. Male and female data that are currently analysed in separate models should be analysed in the same models with sex as a main factor, these analyses should be reported too.

 We recognized that the rationale behind fitting models to male and female data separately was not clearly explained in the original manuscript. Our aim was to characterize how song type (father’s and unfamiliar), age, and their interaction predict the behavior of each sex, under the hypothesis that if sex differences contribute to the ontogeny of song preference, stimulus type, age of testing, and their interaction would predict behavioral response in a different manner depending on sex. We explained these points in the Methods (p. 12, lines 204-206; pp. 14-16, lines 243-284) and Results (p. 19, lines 321-325) sections. Another reason males and females were treated independently was that we wanted to analyze the phonotaxis, calling, and singing responses within the identical framework. To meet this purpose, we decided that applying models to male and female datasets separately would be more appropriate, especially because the sex difference in vocal response could be qualitative rather than quantitative. Although we did not explicitly conduct statistical tests to directly compare the behavioral proportions of males and females, we believe that readers can get equivalent information from the mean value and confidence intervals of the behavioral proportion data (indicated in Figs 2a, 3a, 3d, and Tables 1 and 3).

(continued) Another important point for the analyses I noted: The design consists of the same tests for 10 males and 10 females from 11 different broods. This suggests that there is a male and a female per tutor? If so the design would be much stronger (consisting of paired m/f data belonging to the same father?) In this case tutor ID should be part of the analyses.

We realized that we did not give enough description about the subjects (e.g. the number of subjects per brood). We revised the manuscript and prepared additional Supporting Information to give more details (p. 6, lines 96-98 and S1 Table). Although 2 or 3 birds were recruited from one brood in some cases, we did not systematically design the experiment to always recruit a pair of male and female juveniles per brood. Therefore, we could not include tutor ID in our analysis. 

3) A problem with experimental studies of behavioral development involving repeated testing is that age effects can be difficult to disentangle from learning within the experiment. This is a dilemma for studies of behavioral development. It can only be overcome by having many experimental groups each of which gets only tested at a single age and to compare these data to data from groups that get tested at all ages. I realise that for a species with long development and separate housing required in song learning studies this is not always logistically possible but it is important to acknowledge and discuss that some of the changes could potentially have arisen because the birds have experienced the testing setup up repeatedly.

Thank you for noting an important point. We acknowledged that we must always consider that learning within the experiment due to repeated exposure be confounded with changes that occur developmentally. As you would expect, it was practically difficult for us to design the experiment with distinct groups of birds that are only tested at a single age. Thus, we discussed possible confounding factors and limitations in interpretation in the Discussion section ‘Methodological issues’ (p. 30, lines 495-506).

Specific remarks per line number

> 34-38 at this stage in the intro no taxonomic scope has been given – clearly not all species show learned recognition and not all species have hatchlings –

It was unclear which taxa of animals we were referring to. We changed the sentence to include imprinting in precocial birds as an example (p. 3, line 35).

> 49 these refs are all zf and bf – perhaps to indicate that female preference learning is a more general phenomenon beyond the estrilid finches, perhaps quote a review here to stress this as a general phenomenon in females? And make readers more aware that looking at this phenomenon in males is much rarer and unresolved

 As in your comment, female preference learning is more commonly seen in nature and thus more intensively investigated than male preference learning. However, we do not agree that citing more general references better justifies our attempt to compare the behavior of male and female songbirds. Because song is a sexually dimorphic trait that only males possess, studying male song preference might be different from studying male preference (to signals of opposite sex) in general. However, we agree that showing a contrast of female and male preference learning is important, and we revised the manuscript to explain the contrast more in depth rather than citing new references (pp. 3-4, lines 49-56).

> 59ff The referencing in this section and argument is too sweeping and confusing. Your sentence suggests that all these studies have found age and sex differences whereas quite a few did not – especially in zf behavioural response to tutor song is not necessarily different in males and females e.g. (Clayton 1988, Riebel et al. 2002) whereas Kriengwatana et al (2016)presents a metaanalyses of studies on discrimination learning involving artificial stimuli and artificial grammar – in their study females performed slightly better but confer this to another type of discrimination tasks where males were reported to perform better - (Cynx & Nottebohm 1992) which suggest that outcomes of such study might not have a general m/f pattern but much might depend on the type of test? For this paragraph it would thus help readers if you were more specific about what the studies you quote show (see also cmts 49) line 59-60 should also state what the response will be about – the responses to tutor song are not necessarily predictive of other stimuli and vice versa - so please be specific about what questions these studies investigated.

 Thank you for the comment. We acknowledge that our argument was too general, and we needed to elaborate how we refer to the findings from previous studies. Thus, we specified the question asked and result of each study mentioned (pp. 4-5, lines 60-72).

> 61-63 this is interesting as the only species I am aware of where this has been tested is the zebra finch and here males and females prefer the tutor song (Clayton 1988, Riebel et al 2002)

We did learn that both male and female birds basically acquire preference to their father’s (tutor’s) song, which is in agreement with studies in zebra finches. Our idea was that there might be either quantitative or qualitative differences between sexes or developmental stages that could be elucidated by designing a study with both a longitudinal design and a behavioral measure with ecological validity. If there were differences, they would help further understand why and how birds develop song preference. We hope this point became clearer when responding to your previous comment (to lines 59- in the original version).)

> 81, 83 give cage sizes, day length food etc. – see ARRIVE guidelines (seeKilkenny et al. 2010)

We added cage size descriptions (p. 6, lines 92-93 and 99). Other housing environment information such as day length, food availability, temperature, and humidity were the same across all birds kept in our aviary and experimental space, and is described on pp. 6-7 on lines 103-107 (last paragraph of ‘Animals’) and on p. 8 on lines 131-132 (‘Song preference test - Apparatus’). Relatedly, we also added a sentence that birds had a free access to food and water in the middle compartment of the test cage (p. 8, lines 124-125) as we noticed that it was mentioned only in Fig 1 but not in the original manuscript.

> 83 with how many other birds?

Birds were kept with 7-13 other birds in a single-sex cage for a total of 8-14 birds per cage. We added this information on p. 6 in lines 99-100.

> 86 “ another 29 adult males” the ‘another’ is confusing as so far there has been no mentioning of adult subjects (only 10m/f offspring) and if the fathers are in the sample it is also confusing – better specify how many of these males were the fathers (I suppose 11?) It is also unclear why you use the many extra males? You would need one unfamiliar song per tutor = 11 unfamiliar (I am assuming males/females from the same tutor also get the same unfamiliar song, please specify this). Ideally you would use the other tutors – as this allows checking whether any songs are very attractive in themselves – and there would be no need to have extra males? Please give more detail here how the stimulus pairs were made and whether a brother/sister pair got the same stimuli.

As you pointed out, the description on animals and stimuli was insufficient in the original manuscript. Of 29 adult males we used, 11 birds were the fathers of subjects and 18 were unfamiliar individuals (p. 6, lines 100-102). We modified the description on pp. 8-9 in lines 134-141 (‘Methods – Song preference test – Stimuli’), to reflect that we used songs of different unfamiliar birds between tests at different ages. This means that a subject was exposed to a new unfamiliar song at each age of testing. Also, we randomly chose the unfamiliar song to be presented to each bird at each age. Thus, the father’s song was not always paired with a specific unfamiliar song and that is why we used 18 more birds in addition to the fathers. However, we could not systematically design the stimulus pairs so that father’s song of one subject is always used as an unfamiliar song to another subject. We clarified these points on pp. 8-9, lines 138-141. Please also see our response to the question you raised to line 155 of the original manuscript (below), as this is a related issue. 

> 97 phonotaxis test?

We thought it might be more precise to name it ‘song preference test’ because we measured both phonotaxis and vocal responses to song playbacks. Thus, we kept the heading as it was but added a description of what behavior we tested in the following sentence (p. 7, lines 116-117).

>129 give the settings of the sound meter

We specified that we measured the equivalent continuous A-weighted sound pressure level (p. 10, lines 160-162).

> 150 putting the mother in as company seems wise from a welfare point – but can you be sure that she isn’t also reacting to the songs and thus guiding their offspring’s responses?

As described on p. 11 on lines 181-182 and 186-188, the mother stayed in the test chamber (middle compartment) only during the habituation period (when no stimuli were played back, and birds could not enter the approach zones). During testing, the mother was removed from the cage and the subject was alone. Thus, we believe that there was little possibility that the mother somehow biased the response of subject.

> 155 was this always the same unfamiliar song or different ones?

We used songs recorded from one unfamiliar individual within a test at a certain age but used songs of different unfamiliar individuals between tests at different ages. Thus, each bird was exposed to a new unfamiliar song at each age of testing. Also, we randomly chose the unfamiliar song to be presented to each bird at each age, aiming to avoid pseudo-replication and minimize the possible effect of inherent attractiveness of a particular song. We added this description on p. 8 on lines 136-139.

> 122 Please clarify if this duration for one song or for the song stimulus (consisting of 5 renditions of the song?)

This is the mean duration of a single song (one song rendition). We modified the sentence to be clearer (p. 9, line 152).

> 130 would that be the same amplitude a bird sitting at 12,5 cm away from another bird would experience?

We did not specifically set the sound pressure level to be the same amplitude as a live bird singing. Although the perceived amplitude depends on the environment, we primarily followed the description in a similar study which used Bengalese finches (Dunning et al., 2014; they set the SPL to 70 dB measured 13 cm away from the speaker). Before conducting experiments, we confirmed that the stimuli broadcasted from the speakers did not sound unnatural at least to the experimenter’s ears. Also, few birds appeared scared of or startled by song presentation. We judge that the amplitude was close to that of natural settings, though this judgement is partly retrospective.

> 144 at 40 dph if you put one subject in, what happened to other siblings – did they stay with the father? If the birds were acclimated for 3 days the rest of the brood would get older (or were they not tested? sample size suggests not but please state specifically if that only one male/female were tested per brood?) 

At around 40 dph, other siblings of the subject either stayed with their father or were isolated in a soundproof box prior to acclimation. There were cases where more than 1 bird was recruited from a brood. In those cases, it was impossible to test siblings at exactly the same age due to the nature of the experimental schedule; thus, there was a few days of difference in the age at which siblings were tested. We described this in the revised manuscript ‘Methods – Song preference test – Testing schedule’ (p. 10, lines 165-167) and summarized the number of subjects of each sex from each brood in Supporting Information S1 Table, which was originally not included but should provide the necessary information.

(continued) - Can you also specifically mention whether this procedure was the same for the later trials (3 days acclim + mother before testing)?

We added in the revised manuscript that the procedure (3 days acclimation with mother) also applies to testing at later ages (pp. 10-11, lines 176-177). We also slightly modified Figure 1 as the previous version might not have been clear on this point. The overall time course and the schedule of each test are separated into 2 panels (b) and (c).

> 158 just say 40 playbacks (as being moved in the test chamber once might mean ‘trial’ for most readers, the playbacks are stimulus exposures during a trial)

We replaced the term ‘trials’ with ‘playbacks’ here (p. 12, line 196) and in other locations, as well (e.g. p. 15, line 253).

> 168 most studies measure the time spent rather than the entries (a bird could only enter once and stay there which would be a very strong preference, whereas a very agitated bird could fly back and forth a lot and would have many entrances?) – please comment on why you chose entrances rather than time spent and whether the two measures were correlated or not correlated (in which case they would measure different dimensions of the response?)

We described the reason we initially chose the number of entrances as a behavioral index and how we modified the analysis in our response to your first major comment. Please see pp. 4-5 in this letter.

> 178-180 this is a proportion not a ratio

We replaced the term ‘ratio’ with ‘proportion’ here (p. 13, line 225) and in later passages as well. We also updated the y-axis labels of Figs 2a and 3a, d. 

> 193 and 252 what was the rationale to test males and females separately – your study explicitly sets out to test for sex differences – so the data of males and females have to be in the same model – as is a differences in df and also different models parameters (you are not suing the same final models for males and females as table 1 suggests where bold face AIC is not always for the same model?)

We think this is closely related to your second major comment. Could you please refer to our response to the comment on page 6 of this letter? For the latter part of your comment (the question in parenthesis): In the initial analysis, we applied the identical set of models to both male and female data, then selected the model for each sex which best predicted the behavior. This means that the procedure of model fitting was exactly the same for both sexes and the difference in the selected models is a result to be reported. Thus, the difference in model selection highlights sex differences in behavior, which was the goal of our study. However, we reconsidered how to analyze the data and thought that this kind of model selection might be redundant and not necessarily the most appropriate method. Instead, we decided to apply only one model which included all three explanatory variables (stimulus, age, and interaction) to see how these variables predict the data.

> 228-234 present these data in a table to improve readability of the text and to aid better visual comparison of the many values?

We prepared additional tables (Table 1 and 3) to show these values in a more readable way. (These values are the numerical description of data visualized in Fig 2a, 3a and 3d.)

> References

- Please be consistent in how you use caps in titles

- species names should be in italics

Beecher MD and Brenowitz EA 2005: Functional aspects of song learning in songbirds. Trends Ecol Evol 20: 143-149. 10.1016/j.tree.2005.01.004

Clayton NS 1988: Song discrimination learning in zebra finches. Anim Behav 36: 1016-1024.

Cynx J and Nottebohm F 1992: Role of gender, season, and familiarity in discrimination of conspecific song by zebra finches (taeniopygia guttata). Proc Natl Acad Sci USA 89: 1368-1371.

Kilkenny C et al. 2010: Improving bioscience research reporting: The arrive guidelines for reporting animal research. PLoS Biol 8: e1000412.

Kriengwatana B et al. 2016: Auditory discrimination learning in zebra finches: Effects of sex, early life conditions and stimulus characteristics. Anim Behav 116: 99-112. 10.1016/j.anbehav.2016.03.028

Riebel K et al. 2019: New insights from female bird song: Towards an integrated approach to studying male and female communication roles. Biol Lett 15: 20190059. 10.1098/rsbl.2019.0059

Riebel K et al. 2002: Sexual equality in zebra finch song preference: Evidence for a dissociation between song recognition and production learning. Proceedings of the Royal Society of London Series B - Biological Sciences 269: 729-733.

Ten Cate C et al. 1984: The influence of differences in social experience on the development of species recognition in zebra finch males. Anim Behav 32: 852-860.

Witte K 2006: Time spent with a male is a good indicator of mate preference in female zebra finches. Ethology Ecology & Evolution 18: 195-204.

 We corrected the capitalizations of these references (refs 6, 7, 14, 27, 31, 43, 47, 50, and 59 in the new version of manuscript) and italicized the species names when scientific names are included in the titles (refs 27, 35, 48, and 58).

---

## [Decision Letter · Decision Letter 1]

27 Oct 2020

PONE-D-20-07853R1

Sex differences in the development and expression of a preference for familiar vocal signals in songbirds

PLOS ONE

Dear Dr. Fujii,

Thank you for submitting your manuscript to PLOS ONE. After careful consideration, we feel that it has merit but does not fully meet PLOS ONE’s publication criteria as it currently stands. Therefore, we invite you to submit a revised version of the manuscript that addresses the points raised during the review process.

Your Manuscript is much improved following the revision and both referees commend the efforts. To make the MS still better and more interesting, I would ask for you to consider the complementary analyses suggested.

We look forward to receiving your revised manuscript.

Kind regards,

Nicolas Chaline

Academic Editor

PLOS ONE

Reviewers' comments:

Reviewer's Responses to Questions

**Comments to the Author**

1. If the authors have adequately addressed your comments raised in a previous round of review and you feel that this manuscript is now acceptable for publication, you may indicate that here to bypass the “Comments to the Author” section, enter your conflict of interest statement in the “Confidential to Editor” section, and submit your "Accept" recommendation.

Reviewer #1: All comments have been addressed

Reviewer #2: (No Response)

2. Is the manuscript technically sound, and do the data support the conclusions?

Reviewer #1: Yes

Reviewer #2: Partly

3. Has the statistical analysis been performed appropriately and rigorously? 

Reviewer #1: Yes

Reviewer #2: Yes

4. Have the authors made all data underlying the findings in their manuscript fully available?

Reviewer #1: Yes

Reviewer #2: Yes

5. Is the manuscript presented in an intelligible fashion and written in standard English?

Reviewer #1: Yes

Reviewer #2: Yes

6. Review Comments to the Author

Reviewer #1: I am happy with the thorough revision the authors have carried out and with the new manuscript submitted.

Reviewer #2: The authors have substantially revised the manuscript and have addressed most points either as suggested. The methods are better to understand now and I have but a few follow up questions and suggestions:

1) Analyses of sex differences: this is the main aim of the study and I understand the authors argumentation to run separate models for males and females where response variables are not fully congruent or because they want to explore which behaviour best predicts their response. However, given that birds generally show individual variation and this is a relatively small sample it is adamant that males and females are also tested in a global model for those parameters that could be measured in both sexes, for example the approach time to familiar/unfamiliar song. You see in both the f/m analyses that there is more approach to the familiar stimulus, I do not see why you cannot, as a posthoc analysis now test whether males and females differ (response time is the dependent, sex*age and their interaction as factors? I think such an analysis is crucial to many of your discussion points?

2) Line 243ff Time spent for familiar versus unfamiliar song are mutually exclusive (the time spent in one zone cannot spent be in another). It should therefore be analysed as proportion response time familiar/total response time in approach zones F +UF. This also could simplify your analyses as you have one instead of two response variables..

More specific comments by line number

33 ‘can have’ instead of ‘has’?

43 females learn too (as you show here yourself) and see your statement line 46!

47 m/f learned preference for father’s song has been shown in only a few species; maybe ZF and BF are actually the only ones?

50 give references for ‘some studies have already demonstrated’ already here?

55 what do you mean with ‘attached’ here?

68 what do you mean with strategy in this context?

69 make clear which study is in which species – it reads as if these are all BF studies

135 is this the same song for all subjects or a different song for each subject?

137 chose from how any songs?

147 Could you maybe provide an example (or give a ref here)? This might greatly help - not all readers will know BF song?

159 peak sound pressure level?

209 usage of ‘trial’ unclear here – do you mean per song playback?

210-11 Would counting the calls just after the playback (i.e. in the interval until the new song started?) be also informative of their interest in the song/singer of the song? birds might especially call just after the playback finished?

216-17 but this is what you want to find out?

260 for all parameters or only for song?

298 why are you here using proportion but not in analyses explained l 243 ff?

7. PLOS authors have the option to publish the peer review history of their article (what does this mean?). If published, this will include your full peer review and any attached files.

Reviewer #1: No

Reviewer #2: No

---

## [Author Response · Author response to Decision Letter 1]

8 Nov 2020

We appreciate the reviewers’ and editor’s careful reading and thoughtful comments on our manuscript. We now conducted a complementary analysis on the sex difference of approach behavior as suggested by the editor and reviewer. We also revised the manuscript to clarify some points raised by the reviewer. Specific responses and details about modification are described below (line numbers refer to the latest version of our revised manuscript). We hope that we have correctly addressed all your suggestions, and the analysis will be easier to understand with the supplement and modification.

Editor:

> Your Manuscript is much improved following the revision and both referees commend the efforts. To make the MS still better and more interesting, I would ask for you to consider the complementary analyses suggested.

Thank you for your advice. We now performed the complementary analysis according to the comment by reviewer 2.

Reviewer #1:

>I am happy with the thorough revision the authors have carried out and with the new manuscript submitted.

We thank you very much for your comments which improved the manuscript.

Reviewer #2:

>The authors have substantially revised the manuscript and have addressed most points either as suggested. The methods are better to understand now and I have but a few follow up questions and suggestions:

Thank you for your thorough reading of our revised manuscripts. We additionally modified the manuscript particularly regarding the referencing in the Introduction section and the analysis of sex differences as you pointed out.

1) Analyses of sex differences: this is the main aim of the study and I understand the authors argumentation to run separate models for males and females where response variables are not fully congruent or because they want to explore which behaviour best predicts their response. However, given that birds generally show individual variation and this is a relatively small sample it is adamant that males and females are also tested in a global model for those parameters that could be measured in both sexes, for example the approach time to familiar/unfamiliar song. You see in both the f/m analyses that there is more approach to the familiar stimulus, I do not see why you cannot, as a posthoc analysis now test whether males and females differ (response time is the dependent, sex*age and their interaction as factors? I think such an analysis is crucial to many of your discussion points?

2) Line 243ff Time spent for familiar versus unfamiliar song are mutually exclusive (the time spent in one zone cannot spent be in another). It should therefore be analysed as proportion response time familiar/total response time in approach zones F +UF. This also could simplify your analyses as you have one instead of two response variables.

Thank you again, for an important suggestion. Regarding these two main comments, we understood your concern that data of males and females should be treated in a single model in which sex, age, interaction are the independent variables, given the individual variation and relatively small sample size in the current experiment. We also agree that approach response (time duration) is comparable between sexes and can be analyzed using the response proportion. Therefore, we performed an additional analysis; a linear mixed model which includes sex, age, and their interaction as explanatory variables was applied to data of response proportion of time spent. We included this analysis in the Supporting Information (S4 Text) and referred to the SI in the main text (lines 244-247, line 667).

More specific comments by line number

> 33 ‘can have’ instead of ‘has’?

We replaced ‘has’ with ‘can have’ as it was too assertive (line 34).

> 43 females learn too (as you show here yourself) and see your statement line 46!

We changed the sentence to correctly convey that only males are ‘vocal’ learners (line 43).

> 47 m/f learned preference for father’s song has been shown in only a few species; maybe ZF and BF are actually the only ones?

We now specified that we refer to previous studies that were conducted on zebra finches and Bengalese finches (lines 47-48). We agree that the sentence in previous version told as if it can be generally observed in different species of songbirds.

> 50 give references for ‘some studies have already demonstrated’ already here?

We now gave the references here (in this sentence, we intended to summarize the instances cited in the next sentence). (line 51).

> 55 what do you mean with ‘attached’ here?

We meant the social attachment (measured by responses such as approaching the tutor). We slightly modified the phrase to more correctly convey this point (line 56).

> 68 what do you mean with strategy in this context?

We used the word ‘strategy’ to mean the way a bird perform a given task. As described in the next sentence, there are sometimes individual differences in what cues in the stimuli a bird learns to use. We slightly modified the sentence for clarification (line 69).

> 69 make clear which study is in which species – it reads as if these are all BF studies

The target of this meta-analysis paper (ref. 28) was auditory discrimination learning in zebra finches. We now made it clear (line 72).

>135 is this the same song for all subjects or a different song for each subject?

Different subjects were tested with different unfamiliar song (different ID of unfamiliar singer). We added this information (lines 136-137).

> 137 chose from how many songs?

Since a father’s song for one subject was used, in some cases, as an unfamiliar song for an unrelated subject hatched from another family (lines 140-143), the number of unfamiliar song candidates varied among individuals. For this reason, we cannot specify from how many songs we chose the stimulus, but we think that the number of males (11 fathers and 18 other males, 29 in total) used for stimulus recording in the section of ‘Animals’ can substitute for that.

> 147 Could you maybe provide an example (or give a ref here)? This might greatly help - not all readers will know BF song?

We now gave a reference for sound spectrograms of the Bengalese finch songs (line 149). Accordingly, the order of references (refs 35 and 36) changed (lines 152, 176, 605-608).

> 159 peak sound pressure level?

Peak sound pressure level was adjusted at 70 dB in the experimental environment (line 162). In line 159 in the previous manuscript, we explained the procedure to control for the variability of amplitude in recorded sounds within each singer.

> 209 usage of ‘trial’ unclear here – do you mean per song playback?

We replaced ‘trials’ with ‘playbacks’ (line 211).

> 210-11 Would counting the calls just after the playback (i.e. in the interval until the new song started?) be also informative of their interest in the song/singer of the song? birds might especially call just after the playback finished?

Thank you for your suggestion. We initially counted the number of calls during song presentation as well as stimulus intervals. As you pointed out, some birds emitted calls after the stimulus offset, which could also be interpreted as a response to the song. However, in the design of current experiment, mean stimulus duration was long enough (approx. 7 seconds) for a bird to respond during presentation. In addition, as the duration of interval was also relatively long (approx. 23 seconds), we thought that counting all the calls emitted during presentation + intervals would rather decrease the sensitivity of measurement. Although counting the calls emitted during presentation plus a given seconds (for example 2-3 seconds?) just after the stimulus offset may be an alternative, the setting of this time window would be arbitrary. For these reasons, we finally decide to count the calls emitted during presentation.

> 216-17 but this is what you want to find out?

As we agree that this justification may contradict our purpose to describe how males and female express their song preference, we deleted the sentence (lines 218-219). Ideally, we should also consider if any call type is particularly relevant to male song preference (for example, by examining correlation to the well-established preference indicator such as phonotaxis). To perform that kind of analysis, however, objective categorization of calls is required as a preliminary step. Although we empirically know that male Bengalese finches have several categories of calls, there has been no literatures that we can refer, and establishing steady criteria of classification anew is out of scope. Thus, we did not conduct call type specific analysis in males but agree that this is rather a technical limitation.

> 260 for all parameters or only for song?

We now specified that this applies to all parameters (male approach, female approach, male call, female call, and male song). (line 264)

> 298 why are you here using proportion but not in analyses explained l 243 ff?

For a visualization / description purpose, we thought that summarizing the frequency / duration data as a proportion would be easier to understand. However, as it is difficult to assume a specific probability distribution underlying proportion data, it is not suitable for statistical analysis (although conventionally used). Rather, we considered that using the actual frequency / duration value is more appropriate.

---

## [Editor Report · Decision Letter 2]

27 Nov 2020

Sex differences in the development and expression of a preference for familiar vocal signals in songbirds

PONE-D-20-07853R2

Dear Dr. Fujii,

We’re pleased to inform you that your manuscript has been judged scientifically suitable for publication and will be formally accepted for publication once it meets all outstanding technical requirements.

Kind regards,

Nicolas Chaline

Academic Editor

PLOS ONE
---

## [Editor Report · Acceptance letter]

9 Dec 2020

PONE-D-20-07853R2 

Sex differences in the development and expression of a preference for familiar vocal signals in songbirds 

Dear Dr. Fujii:

I'm pleased to inform you that your manuscript has been deemed suitable for publication in PLOS ONE. Congratulations! Your manuscript is now with our production department. 

Kind regards, 

on behalf of

Professor Nicolas Chaline 

Academic Editor

PLOS ONE